# Axial length to corneal radius of curvature ratio and refractive error in Chinese preschoolers aged 4–6 years: a retrospective cross-sectional study

Tao Tang ,[1,2,3,4,5] Heng Zhao,[1,2,3,4,5] Duanke Liu,[1,2,3,4,5] Xuewei Li,[1,2,3,4,5] Kai Wang ,[1,2,3,4,5] Yan Li,[2,4,5] Mingwei Zhao[1,2,3,4,5]

[1]Institute of Medical Technology, Peking University Health Science Center, Beijing, China
[2]Department of Ophthalmology & Clinical Centre of Optometry, Peking University People's Hospital, Beijing, China
[3]College of Optometry, Peking University Health Science Center, Beijing, China
[4]Eye Disease and Optometry Institute, Peking University People's Hospital, Beijing, China
[5]Beijing Key Laboratory of the Diagnosis and Therapy of Retinal and Choroid Diseases, Beijing, China

**Correspondence to**
Dr Kai Wang;
wang_kai@bjmu.edu.cn and
Dr Yan Li;
13801153660@163.net

## ABSTRACT

**Objective** This study aims to investigate the associations of axial length to corneal radius of curvature (AL/CRC) ratio with refractive error and to determine the effect of AL/CRC ratio on hyperopia reserve and myopia assessment among Chinese preschoolers.

**Methods** This was a retrospective cross-sectional study that evaluated subjects aged 4–6 years. AL and CRC were obtained using a non-contact ocular biometer. Correlation analysis was performed to explore the associations of AL/CRC ratio with spherical equivalent refractive error (SER). The accuracy of AL/CRC ratio for hyperopia reserve and myopia assessment was analysed using cycloplegic refraction by drawing receiver operating characteristic (ROC) curves.

**Results** The analysis included 1024 participants (537 boys, 52.4%). The mean AL/CRC ratios in hyperopes, emmetropes and myopes were 2.90±0.06, 2.95±0.05 and 3.08±0.07, respectively. The SER was found to be more strongly correlated with AL/CRC ratio ($\rho$=−0.66, p<0.001) than either AL or CRC alone ($\rho$=−0.52, p<0.001; $\rho$=−0.03, p=0.33, respectively). AL/CRC was correlated with SER in hyperopes ($\rho$=−0.54, p<0.001), emmetropes ($\rho$=−0.33, p<0.001) and myopes (r=−0.67, p<0.001). For low hyperopia reserve assessment, the area under the ROC curves of AL/CRC ratio was 0.861 (95% CI 0.829 to 0.892), the optimal cut-off value of the AL/CRC ratio was ≥2.955. For myopia assessment, the area under the ROC curves of AL/CRC ratio was 0.954 (95% CI 0.925 to 0.982), the optimal cut-off value of the AL/CRC ratio was ≥2.975.

**Conclusions** The SER showed a better correlation with AL/CRC ratio than either AL or CRC alone, especially in myopes, among children aged 4–6 years. These findings indicate that when cycloplegic refraction is unavailable, AL/CRC ratio could be used as an alternative indicator for identifying low hyperopia reserve and myopia among preschoolers, helping clinicians and parents screen children with low hyperopia reserve before primary school in a timely manner.

## STRENGTHS AND LIMITATIONS OF THIS STUDY

⇒ An advantage of this study was the population-based design, with a large sample size of children aged 4–6 years .

⇒ The hyperopic reserve and myopia were assessed by axial length to corneal radius of curvature (AL/CRC) ratio using receiver operating characteristic curves and area under the curve.

⇒ This study is cross-sectional and does not pinpoint the temporal correlations between refractive error and the AL/CRC ratio.

⇒ The sample size of 4-year-old participants was relatively small, which may result in potential for discrepancy.

## INTRODUCTION

Myopia has become a global public health problem worldwide, especially in East Asian countries such as China, Singapore and South Korea.[1–3] It is predicted that, by 2050, 49.8% of the global population will have myopia and 19.7% will suffer high myopia.[4] It is worth noting that the incidence of myopia has shown a trend towards increasing in those who are younger in recent years. In 2011, the overall prevalence of myopia among children aged 5–14 years in Beijing was 36.7%.[5] In Hong Kong, myopia was 18.3% in children aged 6 years and increased to 61.5% in those aged 12 years in 2018.[6] In France, the myopic rate was 19.6% in children aged 0–9 years in 2013.[7] For most myopic patients, refractive error tends to stabilise in adulthood, while for a significant proportion of the myopic population, refraction continues to progress in adulthood and develops into pathological myopia, resulting in various vision-threatening complications such as retinal detachment, glaucoma and cataracts.[8–10] Early monitoring of children's refractive development and timely control of myopia progression will reduce the risk of myopia-related complications in the future life.

Ocular biometrics including axial length (AL), corneal curvature and lens power are among the most important factors affecting refractive status of the eyes.[11] During infancy

and young childhood, most children have a physiological hyperopia of approximately+2.00 dioptres (D), which is defined as hyperopia reserve.[12 13] Balancing changes in AL and ocular refractive components, including the cornea and the lens, leads to emmetropisation.[14] As children grow older, the refraction of the eye develops, and the hyperopia reserve will gradually decrease. Myopia is an ocular disorder characterised by excessive ocular elongation in relation to total ocular power. It has been widely accepted that the myopia shift in refraction in school-children is due to axial elongation.[15 16] When the rate of axial elongation outpaces the changes in optical power of the cornea and the lens, there is a tendency towards myopia onset.[17] For young children, an early reduction in hyperopic reserve might be a risk factor for future myopia development.[18] Zadnik et al found that children in the USA in grade 1 with less than +0.75 D of hyperopia are at increased risk of myopia formation.[18] Yue et al also reported that the cut point for myopia for grade 1 students was +0.31 D in the Chinese population.[19] Myopia develops fastest at the age of 6–7 years and tends to slow down after 11 years.[20] Thus, to prevent the formation of myopia at an early age, clinicians and parents should pay attention to children with low hyperopia reserve and begin to monitor the development of refractive error before primary school.

Typically, cycloplegic refraction is the gold standard for paediatric refractive error diagnosis,[21–23] especially in younger children,[22 23] largely due to an active accommodation response. However, considering that the side effects of cycloplegia such as allergic conjunctivitis, photophobia and poor near visual acuity,[24–26] cause parents or children to be reluctant to undergo cycloplegic refraction. Furthermore, for clinicians who lack regulatory approval for the use of cycloplegics and some patients who are unable to use cycloplegia, cycloplegic refraction may not be performed in such situations. Effective strategies are needed to examine ocular refractive development and diagnose refractive error in young children. Scheiman et al reported that the AL to corneal radius of curvature (AL/CRC) ratio could be a better marker of myopia progression than AL alone.[27] The correlation between refraction and the AL/CRC ratio was stronger than that between refraction and AL,[27–29] and the AL/CRC ratio could explain the total variance in refractive error better than AL alone.[30 31] Moreover, the measurement of AL and CRC is non-invasive, and it is easily accepted by children.[29] Thus, although cycloplegic refraction cannot be replaced by any other measurements for the diagnosis of refractive error in children, as an alternative, the AL/CRC ratio can be used as an approximated surrogate for refractive error estimation.

Although many studies have addressed the correlation between the AL/CRC ratio and refractive error in school-aged children and adult population,[17 32–35] few population-based studies to date have been performed to focus on the correlation between the AL/CRC ratio and refractive error and evaluate the effect of the AL/ CRC ratio for hyperopia reserve and myopia assessment among preschool children in China. Therefore, in this study, we aimed to report the AL/CRC ratio in a Chinese preschool student population. In addition, we examined the correlation between the AL/CRC ratio and refractive error and evaluated the effectiveness of the AL/CRC ratio for refractive error assessment in Chinese preschoolers aged 4–6 years.

## METHODS

### Setting and participants

The study, consisting of participants who visited the Peking University People's Hospital optometry centre between January 2020 and September 2022 due to refractive error, was conducted to investigate AL/CRC ratio and refractive error. All the subjects underwent a comprehensive ophthalmological examination and 1024 subjects fulfilled the following inclusion criteria were enrolled in the retrospective cross-sectional study: (1) age of 4–6 years; (2) astigmatism and anisometropia of 1.50 D or less; (3) the monocular best corrected visual acuity (VA) of 0.00 logMAR (6/6) or better; (4) the absence of any ocular and systemic diseases, such as cataracts, glaucoma and strabismus and (5) cooperate in completing eye examinations.

### Measures

Refraction and ocular biometry examinations were performed for all subjects. First, measurement of ocular biometric parameters was performed with an optical biometer. AL was measured with non-contact partial-coherence laser interferometry (IOL Master; Carl Zeiss Meditec, Oberkochen, Germany). A corneal topography system (the Sirius, Italy) was used to obtain the mean K reading (Kmean) and anterior chamber depth (ACD). Five valid readings were taken and averaged. The measurements were taken by the same experienced technical staff, and the mean value of these measurements was taken for statistical analysis. The ocular parameters presented here are the mean±SD.

After the biometry measurements, all subjects underwent non-cycloplegic refraction with an autorefractor (KP8800; Topcon, Tokyo, Japan). This instrument generated five reliable readings of refraction in both eyes; the median reading was used for analysis. Then, cycloplegic refraction was performed for each subject with three drops of 1% cyclopentolate, the first two drops were administered 5 min apart, and the third drop was administered 20 min later. Cycloplegia was considered complete if the pupil dilated to 6 mm or more and there was no pupillary reflex. The average of five readings automatically performed by the autorefractor was used for analyses. Slit-lamp examination and fundus examination were performed.

### Definitions

Refraction was defined as spherical equivalent refraction (SER; SER=spherical power+cylinder power/2).

Myopia was defined as SER≤−0.50 D, emmetropia as −0.50 D<SER≤+0.50 D and hyperopia as SER>+0.50 D. Signals from the tear film and retinal pigment epithelium are used by non-contact partial-coherence laser interferometry (IOL Master; Carl Zeiss Meditec, Oberkochen, Germany) for AL measurements. ACD was defined as the distance from the anterior corneal surface to the anterior lens surface. The mean K reading (Kmean=(flat K reading+steep K reading)/2) was the average of the steepest and flattest meridians. The CRC was converted from the Kmean data using the formula CRC=0.3375/Kmean (D)×1000. The AL to CRC ratio (AL/CRC ratio) was defined as the AL divided by the mean CRC. Lens power was calculated using the Bennett-Rabbetts method[36] with unknown lens thickness, using measured values for SER, ACD, CRC and AL. The equation was as follows by the formula of Bennett.

$$LP = \frac{L(S_{cv} + K) - 1000n}{(L - ACD - c_{BR})\left(\frac{ACD + c_{BR}}{1000n}(S_{CV} + K) - 1\right)}$$

In this formula, LP is the lens power using the Bennett-Rabbetts method, L is the AL, S is the SER at the corneal vertex, $S_{CV} = S/(1-0.014S)$, K is the mean corneal power, n=1.336 and $c_{BR}$=2.891±0.778 mm.

## Analysis

Statistical analysis was performed by using the SPSS statistical software package (V.22.0, IBM). Descriptive statistics were calculated. Continuous data are expressed as the mean±SD. As biometric data for the right and left eyes were highly correlated (Pearson correlation coefficient, r=0.91), analyses were performed using data for the right eye only. The AL, CRC, AL/CRC ratio and SER were analysed as continuous variables, and the Kolmogorov-Smirnov test was performed to verify the normality of these variables. To show the correlations between SER and AL, CRC and the AL/CRC ratio, Spearman's correlation analyses were performed. The relationship between the AL/CRC ratio and other variables was assessed using Spearman's correlation. SER, AL, CRC, AL/CRC ratio, ACD and LP were compared between groups using independent sample one-way analysis of variance. Separate linear regression models were constructed with SER as the dependent variable and AL/CRC ratio, AL and CRC as the main covariate, respectively. The sensitivity, specificity, Youden index, true positive rate and false positive rate of the AL/CRC ratio and AL for hyperopia reserve and myopia assessment were analysed. The effectiveness of the AL/CR ratio, CRC and AL for hyperopia reserve and myopia assessment was evaluated using receiver operating characteristic (ROC) curves and the area under the curve (AUC) of the ROC curves. All p values were two sided, and a p<0.05 was indicated statistical significance.

## Patient and public involvement

None.

## RESULTS

### Demographic characteristics

A total of 1024 children (537 boys and 487 girls) aged 4–6 years (mean 5.77±0.47 years) were included in this study. The average ages of the boys and girls were 5.79 and 5.76 years, respectively. There was no significant difference in sex distribution ($\chi^2$=2.441, p=0.118). The mean SER was 0.90±1.11 D, mean AL was 22.59±0.76 mm, mean CRC was 7.74±0.24 mm and mean AL/CR ratio was 2.92±0.08. The detailed demographic characteristics of the subjects are shown in online supplemental table 1.

The histograms of SER, AL, CRC and AL/CRC ratio are shown in online supplemental figure 1. The Kolmogorov-Smirnov test indicated that the frequency distribution was non-normal for SER, AL, CRC and the AL/CRC ratio (all p<0.05). The overall prevalence rates of myopia and high myopia were 7.3% and 0.1%, respectively. The prevalence rates of myopia in subjects aged 4, 5 and 6 years were 4.2% (1/24), 5.4% (10/186) and 7.9% (64/814), respectively.

Table 1 shows the distribution of ocular biometry and the AL/CRC ratio according to the categories of refractive status of subjects. AL was significantly longer (p<0.001) in myopes (23.74±0.65 mm) than in emmetropes (22.86±0.63 mm) and hyperopes (22.42±0.67 mm) (p<0.001). The AL/CRC ratio was also significantly higher in myopes (3.08±0.07) than in emmetropes (2.95±0.05) and hyperopes (2.90±0.06) (p<0.001), while CRC and ACD tended to stabilise in different refractive statuses. Emmetropes have the largest LP, while myopes have the smallest LP. Online supplemental figure 2 shows that AL/CRC ratio correlates with age, and the older the age, the greater the value of AL/CRC ratio.

### Correlation analysis between AL, CRC, AL/CRC ratio and SER

Figure 1 shows the scatter plots between AL, CRC, AL/CRC ratio and SER. The SER showed a better correlation with the AL/CRC ratio (Spearman's correlation coefficient, ρ=−0.66, p<0.001) than either AL (ρ=−0.52, p<0.001) or CRC (ρ=−0.03, p=0.33) (online supplemental table 2). The Spearman's correlation coefficient between SER and the AL/CRC ratio was the highest in myopes (ρ=−0.67, p<0.001), followed by hyperopes (ρ=−0.54, p<0.001) and emmetropes (ρ=−0.33, p<0.001). There was a significant correlation between SER and AL in hyperopes (ρ=−0.41, p<0.001) and myopes (ρ=−0.37, p<0.001), but not in emmetropes (ρ=−0.08, p=0.28). There was no statistical significance for the correlation between SER and CRC in all refractive groups (all p>0.05). Compared with either AL or CRC, the AL/CRC ratio showed better correlation with SER in all refractive groups (table 1).

### Regression analysis between AL, CRC, AL/CRC ratio and SER

Univariate and multiple linear regression analyses were performed in this study (online supplemental table 2). For SER variance, AL explained 34.5%, whereas the AL/CRC ratio explained 64.5% of the data after adjusting for age and sex. Multiple linear regression adjusted for age and sex showed that a 0.1 increase in the AL/CRC ratio

**Table 1** Values of ocular parameters and correlations between AL, CRC, AL/CRC ratio and SER according to refractive status

| Parameters | Hyperopes (n=772) | Emmetropes (n=174) | Myopes (n=78) | P value |
|---|---|---|---|---|
| SER | +1.32±0.73 | +0.18±0.30 | −1.70±1.05 | <0.001 |
| AL | 22.42±0.67 | 22.86±0.63 | 23.74±0.65 | <0.001 |
| CRC | 7.74±0.24 | 7.76±0.26 | 7.72±0.21 | 0.678 |
| AL/CRC | 2.90±0.06 | 2.95±0.05 | 3.08±0.07 | <0.001 |
| ACD | 3.35±0.27 | 3.46±0.23 | 3.30±0.26 | 0.140 |
| LP | 24.18±1.27 | 24.31±1.32 | 22.62±1.28 | <0.001 |
| Spearman's correlation coefficients | | | | |
| SER and AL | −0.41* | −0.08 | −0.37* | |
| SER and CRC | −0.06 | 0.10 | −0.15 | |
| SER and AL/CRC | −0.54* | −0.33* | −0.67* | |

*P<0.01
ACD, anterior chamber depth; AL, axial length; AL/CRC ratio, axial length to corneal radius of curvature ratio; D, dioptre; LP, lens power; SER, spherical equivalent refractive error.

was associated with a −1.19 D change in SER and a 1 mm increase in AL was associated with a −0.87 D change in SER.

### Effectiveness of AL, CRC and the AL/CR ratio for hyperopia reserve assessment and myopia assessment

Taking cycloplegic refraction SER ≤+0.50 D as the standard for diagnosis of the lack of hyperopia reserve,[37] the effectiveness of AL, CRC and AL/CR ratio for myopia assessment were analysed (figure 2A). The ROC curves were drawn using AL, CRC and the AL/CR ratio as the index for hyperopia reserve assessment, and the AUCs of the ROC curves were 0.777 (95% CI 0.741 to 0.812), 0.494 (95% CI 0.450 to 0.539) and 0.861 (95% CI 0.829 to 0.892), respectively. If AL was used for hyperopia reserve assessment, the optimal cut-off point of the ROC curve was ≥22.615 mm, with a specificity, sensitivity, Youden index, true positive rate and false positive rate of 0.785, 0.604, 0.389, 0.604 and 0.215, respectively. If the AL/CR ratio was used for myopia assessment, the optimal cut-off point of the ROC curve was ≥2.955, with a specificity, sensitivity, Youden index, true positive rate and false positive

rate of 0.732, 0.863, 0.595, 0.863 and 0.268, respectively (table 2). When optimal cut-off value of AL/CRC ratio was used to assess the lack of hyperopic reserve, 196 of the 252 participants with low hyperopic reserve were correctly classified using the optimal AL/CRC threshold. Using the optimal cut-off value of AL/CRC ratio, 22.22% (56/252) of participants with a lack of hyperopia reserve were not correctly classified (online supplemental table 3).

Taking cycloplegic refraction SER ≤−0.50 D as the gold standard for diagnosis of myopia,[19] the effectiveness of AL, CRC and AL/CR ratio for myopia assessment were analysed (figure 2B). The ROC curves were drawn using AL, CRC and the AL/CR ratio as the index for myopia assessment, and the AUCs of the ROC curves were 0.898 (95% CI 0.864 to 0.932), 0.493 (95% CI 0.434 to 0.553) and 0.954 (95% CI 0.925 to 0.982), respectively. If AL was used for myopia assessment, the optimal cut-off point of the ROC curve was ≥23.235 mm, with a specificity, sensitivity, Youden index, true positive rate and false positive rate of 0.862, 0.800, 0.662, 0.800 and 0.138, respectively. If the AL/CR ratio was used for myopia assessment, the

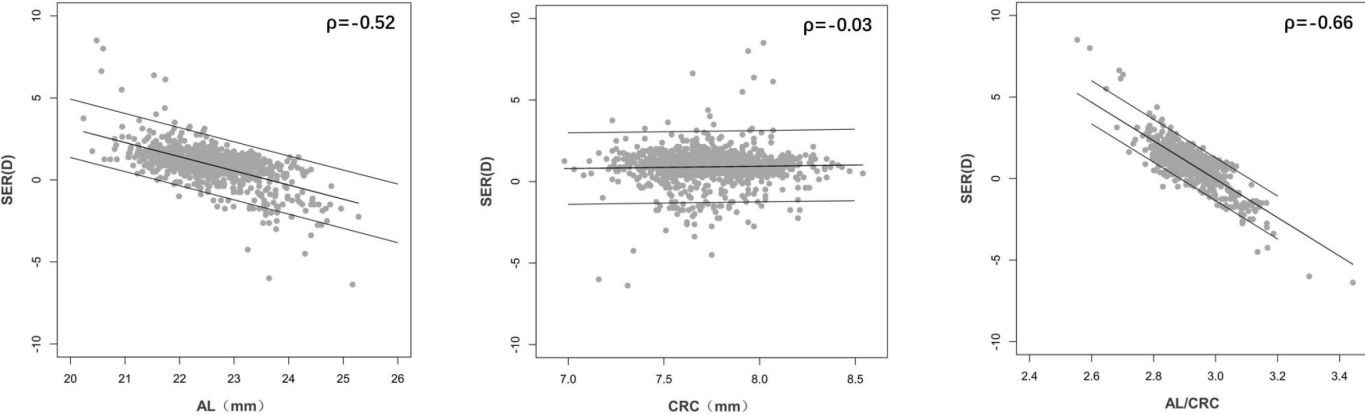

**Figure 1** Correlation analysis between AL, CRC, AL/CRC ratio and SER. AL/CRC ratio, axial length to corneal radius of curvature ratio; D, dioptre; SER, spherical equivalent refractive error.

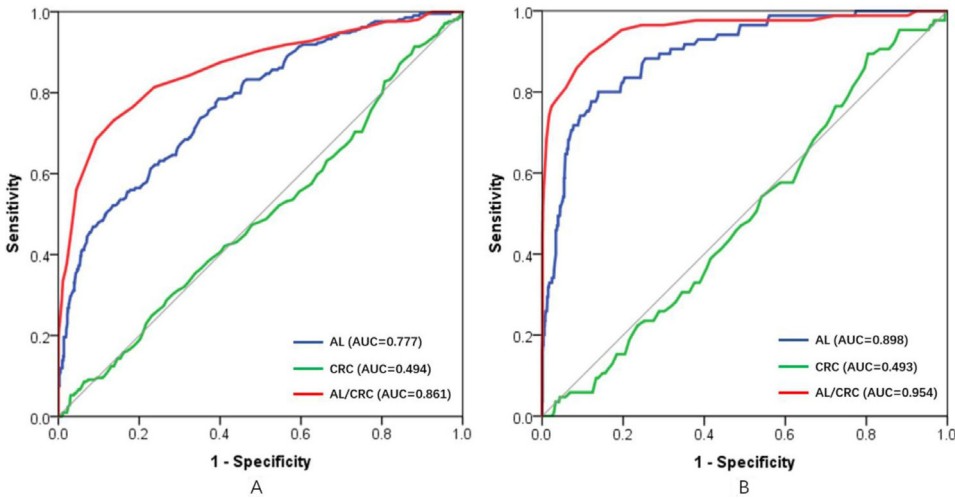

**Figure 2** Comparison of the effectiveness of AL, CRC and the AL/CR ratio for hyperopia reserve assessment (A) and myopia assessment (B). AL, axial length; AUC, area under curve; CRC, corneal radius of curvature.

optimal cut-off point of the ROC curve was ≥2.975, with a specificity, sensitivity, Youden index, true positive rate and false positive rate of 0.883, 0.894, 0.777, 0.894 and 0.117, respectively (table 2). When optimal cut-off value of AL/CRC ratio was used to assess true myopia, 79 of the 85 participants with myopia were correctly classified using the optimal AL/CRC threshold. Using the optimal cut-off value of AL/CRC ratio, 7.06% (6/85) of participants with myopia were not correctly classified (online supplemental table 3).

## DISCUSSION

This study found that AL/CRC ratio was correlated with SER in hyperopes, emmetropes and myopes, while AL showed correlation with SER in hyperopes and myopes, but not in emmetropes. The SER showed a better correlation with the AL/CRC ratio than either AL or CRC alone in different refractive statuses, especially in myopes, among children aged 4–6 years. The effect of the AL/CR ratio for low hyperopia reserve and myopia assessment was higher than that of either AL or CRC alone. These findings indicate that when cycloplegic refraction is unavailable, AL/CRC ratio can be used as an alternative indicator for monitoring low hyperopia reserve and myopia among preschoolers.

Myopia is a common disorder of the eyes and has become a global public health concern.[38 39] During the past several decades, the prevalence of myopia in children has increased rapidly, and the age at onset of myopia has decreased. Chinese children tend to have a higher prevalence of myopia than children in other countries and of other ethnicities. The prevalence of myopia exceeds 60% among 12 years of age in China after primary school, reaches nearly 80% at 16 years of age after junior high school and surpasses 90% in university students.[1 4 13] Notably, the incidence of myopia increased among preschool children in recent years. The prevalence of myopia was 17.6% in children aged 6 years in Hong Kong.[40] A study in Shanghai of 7166 Chinese children aged 4–6 years reported that the prevalence of myopia ranged from 5.7% to 7.1%.[41] In this study, the prevalence of myopia in preschoolers increased from 4.2% at 4 years of age to 7.9% at 6 years of age. Although myopia tends to stabilise after adulthood, children with a younger age at

**Table 2** The sensitivity, specificity and Youden index of AL and the AL/CRC ratio for hyperopia reserve assessment and myopia assessment

| | SER | Criterion | Cut-off value | Sensitivity | Specificity | Youden index |
|---|---|---|---|---|---|---|
| Hyperopia reserve assessment | ≤+0.50 D | AL (mm) | 22.615 | 0.785 | 0.604 | 0.389 |
| | | AL/CRC ratio | 2.955 | 0.732 | 0.863 | 0.595 |
| | ≤+1.00 D | AL (mm) | 22.515 | 0.677 | 0.693 | 0.370 |
| | | AL/CRC ratio | 2.925 | 0.670 | 0.774 | 0.444 |
| | ≤+1.50 D | AL (mm) | 22.495 | 0.625 | 0.778 | 0.403 |
| | | AL/CRC ratio | 2.895 | 0.736 | 0.736 | 0.472 |
| Myopia assessment | ≤−0.50 D | AL (mm) | 23.235 | 0.800 | 0.862 | 0.662 |
| | | AL/CRC ratio | 2.975 | 0.894 | 0.883 | 0.777 |

AL, axial length; AL/CRC ratio, AL to corneal radius of curvature ratio; D, dioptre; SER, spherical equivalent refraction.

myopia onset are prone to develop high myopia and have a high risk of suffering from vision-threatening complications. In view of this condition, it is urgent to carry out early vision screening and refractive error assessment among preschool children. Ocular biometrics have been used to estimate the progression of refractive error.[29 34] The AL/CRC ratio summarises the overall relationship between AL and CRC and minimises variability in AL and CRC, thus, it has been considered as a more robust index to evaluate the refractive status of the eye.[17] Moreover, the AL/CRC ratio also correlates with refractive error more strongly than AL or CRC alone.[28 31] Therefore, the AL/CRC ratio could be useful in determining refractive error and monitoring myopia progression.

Changes in AL and CRC are important ocular biometrics affecting refractive status.[34] Previous studies have indicated that CRC may be actively modulated to regulate refractive development, and the onset of myopia is due to a failure of corneal compensation for axial elongation.[11 33] In this study, we found that there was no correlation between CRC and refractive error, while AL, or more strikingly, the AL/CRC ratio had a significant correlation with refractive error. This finding suggests that AL has a larger effect on refractive error than CRC. Grosvenor was the first to report the correlation between the AL/CRC ratio and refractive error and studied the role of the AL/CRC ratio in determining the refractive state of the eye.[42] They demonstrated that the AL/CRC ratio was the most important biometric factor in myopia. Some researchers further confirmed Grosvenor's findings.[29 43 44] The distribution of AL/CRC ratio has been reported in different countries and age populations. A summary of AL/CRC ratio in other studies is shown in online supplemental table 4. The average values reported for the AL/CRC ratio range from 2.81 to 3.13. On the one hand, differences in age, race, included subjects, measurement methods and techniques may account for the disagreement in the various studies. On the other hand, we can find that the older age, the greater value of AL/CRC ratio. Online supplemental figure 2 also confirms this result. This finding indicates that, in addition to refraction, age also affects the value of AL/CRC ratio. The AL/CRC ratio has been proposed as a calculated biometric parameter that correlates better with SER than AL, and the AL/CRC ratio could explain the total variance in SER better than AL alone.[31] This is because the AL/CRC ratio provides a rough measure of the degree of matching between AL and corneal power. In this study, we also found a better correlation between the AL/CRC ratio and SER in children aged 4–6 years compared with that of AL or CRC alone, especially in children with myopia. It is possibly suggested that the AL/CRC ratio is a more useful marker of progress towards myopic refractions than absolute ocular biometric values such as AL or CRC. Moreover, the value of the AL/CRC ratio was negatively correlated with SER. As demonstrated in this study, the AL/CRC ratio was significantly higher in myopes than in emmetropes and hyperopes, which was consistent with studies performed by Foo et al and Guo et al.[28 31] This finding indicates that the greater the myopic refraction, the larger the AL/CRC ratio.

A previous study reported that a high AL/CRC ratio of no less than 3.0 has been considered an indicator for the onset of myopia.[42] High AL to corneal radius ratio is a risk factor in the development of myopia. When the AL/CRC ratio was higher than 3.0, it was considered that other ocular parameters were not able to compensate for the change in refraction caused by the elongation of AL, thereby inducing myopia. As shown in this study, the area under the ROC of the AL/CR ratio for myopia assessment was 0.954, and the optimal cut-off value is ≥2.975, which was smaller than that in the study performed by Mu et al.[29] This result indicates that for children aged 4–6 years, the risk of myopia is likely to occur when the AL/CRC ratio is greater than 2.975. However, it is worth noting that this AL/CRC threshold is still subject to some degree of error assessment in the evaluation of true myopia. We also conducted an evaluation of the effectiveness of the AL/CRC ratio, CRC and AL for hyperopia reserve assessment in this population. Previous study has reported that hyperopia reserve is a potential indicator of myopia.[19] There is a progressive shift in mean refraction from roughly +2.00 D at 3 months of age to approximately +1.00 D at 6.5 years of age.[45] Then emmetropisation continues at a slower rate after 6 years of age. Zadnik et al revealed that children in the USA in grade 1 with less than +0.75 D of hyperopia are at increased risk of myopia onset.[18] Yue et al also reported that the cut point for myopia for grade 1 students was +0.31 D in the Chinese population.[19] Li et al's[41] findings showed that the 5-year cumulative incidence of myopia was 4.6%, 26.3%, 52.3%, 78.6%, 92.6% and 94.3%, respectively, corresponding to students with a baseline hyperopia reserve of >+2.00 D, +1.50 D to +2.00 D, +1.00 D to +1.50 D, +0.50 D to +1.00 D, 0.00 D to +0.50 D and −0.50 D to 0.00 D, respectively. The lower the hyperopia reserve in young children, the higher the incidence of myopia onset. In our study, taking cycloplegic refraction SER ≤+0.50 D as the standard for diagnosis of the lack of hyperopia reserve, the area under the ROC curve of the AL/CR ratio for the lack of hyperopia reserve assessment was greater than that of AL and CRC and the optimal cut-off value of the AL/CRC ratio was ≥2.955. Although this optimal AL/CRC threshold may have error assessment for preschoolers, clinicians and parents could monitor the hyperopia reserve of young children by regularly measuring the AL and corneal curvature when cycloplegic refraction is unavailable. When the AL/CRC ratio is greater than 2.955 among preschool children, clinicians and parents should pay attention to the children with less hyperopia reserve and begin to take measures to prevent myopia formation.

In this study, AL alone contributed 34.5% of the variance in SER, whereas the AL/CR ratio accounted for 64.5%. Guo et al study[31] found that AL alone contributed 18.6% of the variance in SER, whereas the AL/CR ratio accounted for 39.8% in the study population

of preschoolers aged 3–6 years. He *et al*[30] reported that AL/CR ratio and AL explained 66.4% and 43.1% of the variance in SER, respectively, in a group of Chinese children 6–12 years of age. This suggested that the AL/CRC ratio is more closely related to refractive error than AL or CRC alone. Moreover, we found a linear increase in AL/CRC ratio from hyperopia towards myopia. The corresponding increase in SER of −1.1 9 D with every 0.1 increase in the AL/CRC ratio in this study is larger than −0.74 D reported by researchers in Singapore[28] and −0.89 D reported among the Nigerian population,[33] but slightly lower than −1.21 D reported among northern Iranian subjects.[17] The above findings revealed that in addition to AL and CRC, other potential factors influence SER results. Thus, the AL/CRC ratio cannot completely replace cycloplegic refraction in determining refraction in children. However, cycloplegic refraction is easily influenced by the cooperation and compliance of the examinees, especially in younger children, which is not feasible for large-scale myopia assessment and long-term refractive development follow-up. For this reason, the AL/CRC ratio could be used as an alternative indicator for refraction estimation, and it may be more accurate than AL. In addition, measurement of AL and CRC and determination of the AL/CRC ratio are relatively easy and can be conducted by a technician using non-contact, objective and automated instruments. The application of the AL/CRC ratio reference as an indicator of refractive errors has the effect of simplifying the examination procedure and reducing unnecessary cycloplegia, especially when cycloplegic refraction is difficult to perform in children aged 4–6 years.

There were some limitations in this study. First, because this study is cross-sectional and does not pinpoint the temporal correlations between refractive error and the AL/CRC ratio, it is impossible to rule out the possibility of observational and inclusionary biases due to the retrospective study design. Admittedly, longitudinal studies would be needed to evaluate intraindividual discrepancy. In addition, the data from Beijing may not be representative for the rural areas and other regions in China. Last, the sample size of the 4-year-old participants (N=24) was relatively small. Although our findings are applicable to this population as well, there is potential for discrepancy. We will further increase the sample size of this population to increase the accuracy and reliability of the results of this study.

In conclusion, our study found that SER showed a better correlation with the AL/CRC ratio than either AL or CRC alone in different refractive statuses, especially in myopic children, among Chinese children aged 4–6 years. The AL/CRC ratio could explain the total variance in refractive error better than AL alone. The effect of the AL/CR ratio for assessing the lack of hyperopia reserve and for the myopia assessment was higher than that of AL and CRC. These findings indicate that when cycloplegic refraction is unavailable, the AL/CRC ratio could be used as an alternative indicator for identifying low hyperopia reserve and myopia among preschoolers, helping clinicians and parents screen children with low hyperopia reserve before primary school in a timely manner. In the future, longitudinal cohort designs will be conducted for optimal study of the effect of the AL/CRC ratio on hyperopia reserve and myopia assessment in preschool children.

**Contributors** TT developed the concept, collected data, performed the experiments and data analysis, wrote the main manuscript text and prepared figures 1–2, tables 1–2, online supplemental figures 1–2, online supplemental tables 1–4; HZ collected data and prepared figure 2; DL and XL collected data; KW, YL developed the concept and designed the experiments; MZ developed the study conception. TT wrote the first draft of the manuscript. TT, KW and YL made critical revisions. All authors reviewed and approved the final draft. TT is the guarantor and responsible for the overall content.

**Funding** This work was supported by: National Natural Science Foundation of China (Grant Nos. 82371087 and 82171092), Capital's Funds for Health Improvement and Research (No. 2022-1G-4083), the National Key R&D Program of China (No.2020YFC2008200 and No.2021YFC2702100).

**Competing interests** None declared.

**Patient and public involvement** Patients and/or the public were not involved in the design, or conduct, or reporting, or dissemination plans of this research.

**Patient consent for publication** Not applicable.

**Ethics approval** This study involves human participants and this study was approved by the institutional research ethics committee of Peking University People's Hospital (2021PHB322-001). Participants gave informed consent to participate in the study before taking part.

**Provenance and peer review** Not commissioned; externally peer reviewed.

**Data availability statement** Data are available on reasonable request.

**ORCID iDs**
Tao Tang http://orcid.org/0000-0003-2876-9543
Kai Wang http://orcid.org/0000-0002-4171-825X

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
