## [Reviewer comments · BMJ Open]

ARTICLE DETAILS

TITLE (PROVISIONAL)	Axial length to corneal radius of curvature ratio and refractive error in 4- to 6-year-old Chinese preschoolers: a retrospective cross-sectional study
AUTHORS	Tang, Tao; Zhao, Heng; Liu, Duanke; Li, Xuewei; Wang, Kai; Li, Yan; Zhao, Mingwei

VERSION 1 – REVIEW

REVIEWER	Kiziltoprak, Hasan Ankara Ulucanlar Eye Training and Research Hospital
REVIEW RETURNED	05-Jun-2023

GENERAL COMMENTS	In this article, the authors evaluate the associations of axial length to corneal radius of curvature (AL/CRC) ratio with refractive error and to determine the effect of AL/CR ratio on hyperopia reserve and myopia assessment among Chinese preschoolers. Although the authors explain the purpose and results of the study in a good way in the article, we see the existence of similar studies in the literature. The number of patients included in the study can be increased and the prospective design of the study can bring more positive results. Despite all this, I think it would be more appropriate to accept the article.
--

REVIEWER	Chen, Yanyan Wenzhou Medical University
REVIEW RETURNED	09-Aug-2023

GENERAL COMMENTS	This study aims to investigate the associations of axial length to corneal radius of curvature (AL/CRC) ratio with refractive error and to determine the effect of AL/CRC ratio on hyperopia reserve and myopia assessment among Chinese preschoolers. This is an interesting research question and I think the manuscript has potential, but there are concerns that the authors need to address. 1) In this paper, the age distribution of children is uneven, there are only 24 4-year-old children, and all of them are children who come to the hospital, so the sample is not representative enough. It is suggested to supplement the relevant data on 4-year-olds, or the article focuses on 5-6-year-old children as preschool representatives.2) Cross-sectional studies are not convincing, and it is recommended to add follow-up data from these populations to analyze the association between AL/CRC and myopia progression.3) Some references need to be updated, and it is suggested that they be replaced by 3-5 years.
--

REVIEWER	Zhao, Yune Wenzhou Medical University Eye Hospital
REVIEW RETURNED	10-Sep-2023

GENERAL COMMENTS	The manuscript “Axial length to corneal radius of curvature ratio and refractive error in 4- to 6-year-old Chinese preschoolers: a cross-sectional study” investigated the associations of axial length to corneal radius of curvature (AL/CRC) ratio with refractive error and to determine the effect of AL/CRC ratio on hyperopia reserve and myopia assessment among Chinese preschoolers. This is a meaningful supplement. I only have some minor comments. As anterior chamber depth (ACD) was measured with IOL Master and a corneal topography system (the Sirius, Italy) , which value did the authors used? I believe tihs is a cross-sectional study, please revise “retrospective study in line 153”. Write PL,BR in line-176, LP in line-185, is a same concept? Please write it clearly.
---

REVIEWER	Sheng Bi, Hong Shandong University, ye Institute of Shandong University of Traditional Chinese Medicine
REVIEW RETURNED	18-Sep-2023

GENERAL COMMENTS	The aim of the cross-sectional study is to investigate the associations of AL/CRC ratio with refractive error and to determine the effect of AL/CRC ratio on hyperopia reserve and myopia assessment among Chinese preschoolers. It is of great significance for myopia control in children, especially for screening and early myopic prevention. There were two myopic definitions in this study (Line 169 and Line 238). In order to better compare with other studies in the future, it is recommended to define myopia according to the IMI standard, where SE is less than or equal to -0.5D as myopia. We all know that cycloplegic refraction should be conducted in determining refractive error in children. The authors also mentioned that “AL/CRC ratio cannot replace cycloplegic refraction in determining refractive error in children”. Therefore, cycloplegic refraction should continue to be encouraged, even though AL/CRC has higher accuracy in predicting the refractive index of cycloplegia than AL alone. I suggest the authors to further emphasize that AL/CRC cannot completely replace cycloplegic refraction, in order to avoid misunderstandings among parents and non-professionals. The authors mentioned in the limitations “the sample size of the 4-year-old participants (N=24) was relatively small”. When comparing the prevalence of myopia, consideration should be given to sampling standards and myopia definition standards (Line 260-263). The authors took cycloplegic refraction $SER \leq +0.50$ D as the standard for diagnosis of the lack of hyperopia reserve. For children less than 6 years old, we hope they have more hyperopia
--

	reserve, so that they will not experience myopia or later in the future. Therefore, the authors can consider adding evaluation cut-off values with higher hyperopia reserve values (such as SER $\leq 1.0D$) to Table 3, in order to provide reference for clinical practice or screening. At last, I would like to thank our colleague Dr. Yuanyuan Hu for her suggestions on this review.
--	--

REVIEWER	Lingham, Gareth Technological University Dublin
REVIEW RETURNED	20-Sep-2023

GENERAL COMMENTS	The authors present a clear, well-written retrospective, clinic-based, cross-sectional study of 1,024 4-6 year-old children, in which they investigate the use of the axial length to corneal radius of curvature (AL/CRC) ratio as a marker of spherical equivalent refractive error and as a screening tool for identifying children with myopia or a low hyperopic reserve (and hence are at risk of myopia onset). The methods and statistical analysis are appropriate to answer these questions or limitations are otherwise acknowledged in the discussion. In terms of novelty, the AL/CRC has been well established as a marker of spherical equivalent refractive error and has been investigated as a tool for identifying myopia in children; the main novelty of this study, therefore, lies in the use of AL/CRC to identify children at high risk of myopia onset. This has application in pre-school or school-age eye screening programs where cycloplegia is too invasive and time-consuming and uncorrected visual acuity assessment will not detect imminent myopia. Major comments: As the main novelty of this study lies in the use of the AL/CRC as a screening tool to detect myopia or low hyperopic reserve, I suggest the authors ensure there is adequate focus on this topic and my comments revolve around this.  • I suggest the authors consider including a table that shows the true myopia/low hyperopic reserve classification and the AL/CRC myopia/low hyperopic reserve classification, using the optimal thresholds. This will allow a greater appreciation of the proportion of myopia and low hyperopia that will be missed – often a major concern of clinicians in classification studies such as this. • Pre-school eye screening exams often use uncorrected visual acuity or presenting visual acuity to identify children with myopia. It would be worthwhile comparing the AL/CRC classification to classifications from uncorrected/presenting visual acuity, for both myopia and low hyperopic reserve. This allows readers to compare the AL/CRC to an existing screening classification test and will highlight the added ability of AL/CRC to identify children at high risk of myopia onset (who will have normal visual acuity). • Supplementary Table 2 presents an interesting comparison of AL/CRC across different studies. It is worth adding a measure of spread of AL/CRC to this (e.g. standard deviation). Of more interest is that there appears to be an age-related increase in AL/CRC across studies. Could the author investigate this relationship further and consider quantifying (this could become part of the results). For example, by plotting mean AL/CRC over mean age – it looks to maybe have a logarithmic relationship. This would further indicate if the differences between studies are entirely driven by age or if race, methods etc, are indeed impact
--

	the cohort's mean AL/CRC. This adds to the novelty of the current study. Minor comments:  • Line 127: "orthopedic" is not the correct word to use here • Could the authors elaborate further on why the ages of participants are skewed so heavily toward the 6-year-olds? Is this to do with the age they present to the clinic or parents of younger children refusing participation or other reason? • Conclusions of abstract – I think more uncertainty should be added to the final sentence something like "AL/CRC ratio could be used as an alternative indicator for identifying low hyperopia reserve and myopia. • References to myopia progression in the abstract and manuscript conclusion should be removed here as this study has not assessed this aspect.
--	---

VERSION 1 – AUTHOR RESPONSE

Reviewer #1:

1. In this article, the authors evaluate the associations of axial length to corneal radius of curvature (AL/CRC) ratio with refractive error and to determine the effect of AL/CR ratio on hyperopia reserve and myopia assessment among Chinese preschoolers.

Although the authors explain the purpose and results of the study in a good way in the article, we see the existence of similar studies in the literature. The number of patients included in the study can be increased and the prospective design of the study can bring more positive results. Despite all this, I think it would be more appropriate to accept the article.

Thank you for your comment. In further study, we will increase the sample size of this population and conduct a prospective design to increase the accuracy and reliability of the results of this study.

Reviewer #2:

This study aims to investigate the associations of axial length to corneal radius of curvature (AL/CRC) ratio with refractive error and to determine the effect of AL/CRC ratio on hyperopia reserve and myopia assessment among Chinese preschoolers. This is an interesting research question and I think the manuscript has potential, but there are concerns that the authors need to address.

1. In this paper, the age distribution of children is uneven, there are only 24 4-year-old children, and all of them are children who come to the hospital, so the sample is not representative enough. It is suggested to supplement the relevant data on 4-year-olds, or the article focuses on 5-6-year-old children as preschool representatives.

Thank you for your comment. The sample size of the 4-year-old participants (N=24) was relatively small. Although our findings are applicable to this population as well, there is potential for discrepancy. Therefore, in further study, we will increase the sample size of this population (4-year-olds) to increase the accuracy and reliability of the results of this study.

2. Cross-sectional studies are not convincing, and it is recommended to add follow-up data from these populations to analyze the association between AL/CRC and myopia progression.

Thank you for your comment. In further study, we will add follow-up data from these populations to analyze the association between AL/CRC ratio and myopia progression and conduct a prospective design to increase the accuracy and reliability of the results of this study.

3. Some references need to be updated, and it is suggested that they be replaced by 3-5 years.

Thank you for your comment. We have replaced some references (9, 16, 44) in recent 3-5 years.

Reviewer #3:

The manuscript "Axial length to corneal radius of curvature ratio and refractive error in 4- to 6-year-old Chinese preschoolers: a cross-sectional study" investigated the associations of axial length to corneal radius of curvature (AL/CRC) ratio with refractive error and to determine the effect of AL/CRC ratio on hyperopia reserve and myopia assessment among Chinese preschoolers. This is a meaningful supplement. I only have some minor comments.

1. As anterior chamber depth (ACD) was measured with IOL Master and a corneal topography system (the Sirius, Italy), which value did the authors used?

Thank you for your comment. Anterior chamber depth (ACD) was measured with a corneal topography system (the Sirius, Italy) in this study. We have revised in Page 6 Line 151.

2. I believe this is a cross-sectional study, please revise "retrospective study in line 153".

Thank you for your comment. We have revised in Page 6 Line 144.

3. Write PL,BR in line-176, LP in line-185, is a same concept? Please write it clearly.

Thank you for your comment. PL,BR in line 176 is the lens power (LP) using the Bennett-Rabbetts method. We have revised PL,BR to PL in Page 7 Lines 167-168.

Reviewer #4:

The aim of the cross-sectional study is to investigate the associations of AL/CRC ratio with refractive error and to determine the effect of AL/CRC ratio on hyperopia reserve and myopia assessment among Chinese preschoolers. It is of great significance for myopia control in children, especially for screening and early myopic prevention.

1. There were two myopic definitions in this study (Line 169 and Line 238). In order to better compare with other studies in the future, it is recommended to define myopia according to the IMI standard, where SE is less than or equal to -0.5D as myopia.

Thank you for your comment. We have revised as follows:

Page 7 Line 160

"Myopia was defined as $SE \leq -0.50$ D".

2. We all know that cycloplegic refraction should be conducted in determining refractive error in children. The authors also mentioned that "AL/CRC ratio cannot replace cycloplegic refraction in determining refractive error in children". Therefore, cycloplegic refraction should continue to be encouraged, even though AL/CRC has higher accuracy in predicting the refractive index of cycloplegia than AL alone. I suggest the authors to further emphasize that AL/CRC cannot completely replace cycloplegic refraction, in order to avoid misunderstandings among parents and non-professionals.

Thank you for your comment. We have revised as follows:

Page 13 Lines 293-294

"Thus, the AL/CRC ratio cannot completely replace cycloplegic refraction in determining refractive error in children, avoiding misunderstandings for parents and non-professionals."

3. The authors mentioned in the limitations "the sample size of the 4-year-old participants (N=24) was relatively small". When comparing the prevalence of myopia, consideration should be given to sampling standards and myopia definition standards (Line 260-263).

Thank you for your comment. Indeed, the issue of small sample size the 4-year-old participants (N=24) can have some impact on the prevalence of myopia, and therefore, we will further increase the sample size of this population in subsequent studies to reduce potential discrepancy.

4. The authors took cycloplegic refraction $SER \leq +0.50$ D as the standard for diagnosis of the lack of hyperopia reserve. For children less than 6 years old, we hope they have more hyperopia reserve, so that they will not experience myopia or later in the future. Therefore, the authors can consider adding evaluation cut-off values with higher hyperopia reserve values (such as $SER \leq 1.0D$) to Table 3, in order to provide reference for clinical practice or screening.

Thank you for your comment. We have updated Table 3 in Pages 10-11 Lines 226-227.

Reviewer #4:

The authors present a clear, well-written retrospective, clinic-based, cross-sectional study of 1,024 4-6-year-old children, in which they investigate the use of the axial length to corneal radius of curvature (AL/CRC) ratio as a marker of spherical equivalent refractive error and as a screening tool for identifying children with myopia or a low hyperopic reserve (and hence are at risk of myopia onset). The methods and statistical analysis are appropriate to answer these questions or limitations are otherwise acknowledged in the discussion.

In terms of novelty, the AL/CRC has been well established as a marker of spherical equivalent refractive error and has been investigated as a tool for identifying myopia in children; the main novelty of this study, therefore, lies in the use of AL/CRC to identify children at high risk of myopia onset. This has application in pre-school or school-age eye screening programs where cycloplegia is too invasive and time-consuming and uncorrected visual acuity assessment will not detect imminent myopia.

1. I suggest the authors consider including a table that shows the true myopia/low hyperopic reserve classification and the AL/CRC myopia/low hyperopic reserve classification, using the optimal thresholds. This will allow a greater appreciation of the proportion of myopia and low hyperopia that will be missed – often a major concern of clinicians in classification studies such as this.

Thank you for your comment. We have added Supplemental Table 2 to evaluate effect of optimal cut-off value of AL/CRC ratio for low hyperopia reserve and true myopia assessment and revised as follows:

Page 10 Line 224

“Error low hyperopia reserve assessment using optimal cut-off value of AL/CRC ratio was 22.22% (56/252) (Supplemental Table 2).”

Page 11 Lines 234-235.

“Error myopia assessment using optimal cut-off value of AL/CRC ratio was 7.06% (6/85) (Supplemental Table 2).”

2. Pre-school eye screening exams often use uncorrected visual acuity or presenting visual acuity to identify children with myopia. It would be worthwhile comparing the AL/CRC classification to classifications from uncorrected/presenting visual acuity, for both myopia and low hyperopic reserve. This allows readers to compare the AL/CRC to an existing screening classification test and will highlight the added ability of AL/CRC to identify children at high risk of myopia onset (who will have normal visual acuity).

Thank you for your comment. We believe this is a good suggestion. Since the data of uncorrected visual acuity or presenting visual acuity were not routinely recorded in the files, it was not included in this study for analysis. In subsequent studies, we will further investigate the relationship among AL/CRC ratio, uncorrected visual acuity/presenting visual acuity and myopia/low hyperopic reserve to improve the ability of AL/CRC to identify children at high risk of myopia onset.

3. Supplementary Table 2 presents an interesting comparison of AL/CRC across different studies. It is worth adding a measure of spread of AL/CRC to this (e.g. standard deviation).

Thank you for your comment. Since the references cited in the Supplementary Table 2 does not provide the standard deviation of AL/CRC ratio, we regret that we are unable to provide you with this section.

Of more interest is that there appears to be an age-related increase in AL/CRC across studies. Could the author investigate this relationship further and consider quantifying (this could become part of the

results). For example, by plotting mean AL/CRC over mean age – it looks to maybe have a logarithmic relationship. This would further indicate if the differences between studies are entirely driven by age or if race, methods etc, are indeed impact the cohort’s mean AL/CRC. This adds to the novelty of the current study.

Thank you for your comment. We have revised in Page 12 Line 263 and in Supplemental Figure 2

4. Line 127: “orthopedic” is not the correct word to use here

Thank you for your comment. We have revised as follows:

Page 5 Line 118

“As children grow older, the refraction of the eye develops”.

5. Could the authors elaborate further on why the ages of participants are skewed so heavily toward the 6-year-olds? Is this to do with the age they present to the clinic or parents of younger children refusing participation or other reason?

Thank you for your comment. The reason why the ages of participants are skewed so heavily toward the 6-year-old is that on the one hand, it may be related to the fact that parents do not pay much attention to the refractive developmental status of younger children. On the other hand, 4-5-year-old children are less cooperative with eye examinations, while children over 6 years old cooperate with eye examinations.

6. Conclusions of abstract – I think more uncertainty should be added to the final sentence something like “AL/CRC ratio could be used as an alternative indicator for identifying low hyperopia reserve and myopia.

Thank you for your comment. We have revised as follows:

Page 4 Line 95-96

“AL/CRC ratio could be used as an alternative indicator for identifying low hyperopia reserve and myopia among preschoolers”.

7. References to myopia progression in the abstract and manuscript conclusion should be removed here as this study has not assessed this aspect.

Thank you for your comment. We have removed refers to myopia progression in the abstract and manuscript conclusion.

VERSION 2 – REVIEW

REVIEWER	Lingham, Gareth Technological University Dublin
REVIEW RETURNED	06-Nov-2023

GENERAL COMMENTS	I thank the authors for their time in responding to my comments. However, I still have a few comments or suggestions for improvement remaining.  • Line 294, “Thus, the AL/CRC ratio cannot completely replace cycloplegic refraction in determining refraction in children, avoiding misunderstandings for parents and non-professionals.” I think this sentence is a good addition, but the second part does not make sense. • I think Supplemental Table 2 makes a nice, clinically meaningful addition. However, the newly added results require some extra explanation through table footnotes and in the results and discussion text.
--

	For instance this following sentence is not clear enough to understand. Line 224 “Error low hyperopia reserve assessment using optimal cut-off value of AL/CRC ratio was 22.22% (56/252) (Supplemental Table 2).” Error low hyperopia reserve is not a defined or a commonly used term. From Supplemental Table 2, it looks like there were 252 participants with low hyperopia reserve, of which 196 (77.8%) were correctly classified using the optimal AL/CR threshold. Is this interpretation correct? Again, this needs to be made clear.  • In your response Review #4, comment 5, you state “On the other hand, 4-5-year-olds are less cooperative with eye examinations while children over 6 years cooperate with eye examinations.” Lack of cooperation or of complete data is not listed as an exclusion criteria though, is it the case that children were excluded from the retrospective study if refraction or biometry data were incomplete? This is important to state as it will mean non-cooperative children would be preferentially excluded from your sample. I understand this is unavoidable in this age group but nevertheless the reader should be made aware of it. It would also help to state how many children were excluded for this reason so we know what the potential impact is on sample selection. • Line 263, the reference to Supplemental Figure 2 first needs to occur in the results section. The discussion should spell out the implications of this finding that average AL/CRC changes with age. Presumably your threshold in this study is not going to be applicable to other age groups. • Supplemental Figure 2, thank you for adding this in and it does show that age appears to affect the average AL/CRC. My comment related to plotting the means data shown in Supplemental Table 3 which I think would more clearly show the age vs AL/CR relationship (given the larger age range), but I am satisfied with the authors current Supplemental Figure 2 if that is their preference.
--	--

VERSION 2 – AUTHOR RESPONSE

Reviewer #5:

I thank the authors for their time in responding to my comments. However, I still have a few comments or suggestions for improvement remaining.

1. Line 294, “Thus, the AL/CRC ratio cannot completely replace cycloplegic refraction in determining refraction in children, avoiding misunderstandings for parents and non-professionals.” I think this sentence is a good addition, but the second part does not make sense.

Thank you for your comment. We have revised as follows:

Pages 13-14 Lines 302-303.

“Thus, the AL/CRC ratio cannot completely replace cycloplegic refraction in determining refraction in children.”

2. I think Supplemental Table 2 makes a nice, clinically meaningful addition. However, the newly added results require some extra explanation through table footnotes and in the results and discussion text. For instance, this following sentence is not clear enough to understand. Line 224 “Error low hyperopia reserve assessment using optimal cut-off value of AL/CRC ratio was 22.22% (56/252) (Supplemental Table 2).” Error low hyperopia reserve is not a defined or a commonly used term. From Supplemental Table 2, it looks like there were 252 participants with low hyperopia reserve, of which 196 (77.8%) were correctly classified using the optimal AL/CR threshold. Is this interpretation correct? Again, this needs to be made clear.

Thank you for your comment. We have revised as follows:

The result section:

Page 10 Lines 228-230.

“When optimal cut-off value of AL/CRC ratio was used to assess the lack of hyperopic reserve, 196 of the 252 participants with low hyperopic reserve were correctly classified using the optimal AL/CR threshold. Error lack of hyperopia reserve assessment using optimal cut-off value of AL/CRC ratio was 22.22% (56/252) (Supplemental Table 2).”

The discussion section:

Page 13 Lines 292-293.

“Although this optimal AL/CRC threshold may have error assessment for preschoolers, clinicians and parents could monitor the hyperopia reserve of young children by regularly measuring the axial length and corneal curvature when cycloplegic refraction is unavailable.”

The supplemental Table 2 footnote:

“Error assessment (%), using optimal cut-off value of AL/CRC ratio to assess the probability of errors in lack of hyperopia reserve or true myopia.”

3. In your response Review #4, comment 5, you state “On the other hand, 4-5-year-olds are less cooperative with eye examinations while children over 6 years cooperate with eye examinations.” Lack of cooperation or of complete data is not listed as an exclusion criteria though, is it the case that children were excluded from the retrospective study if refraction or biometry data were incomplete? This is important to state as it will mean non-cooperative children would be preferentially excluded from your sample. I understand this is unavoidable in this age group but nevertheless the reader should be made aware of it. It would also help to state how many children were excluded for this reason so we know what the potential impact is on sample selection.

Thank you for your comment. We have revised as follows:

Page 6 Line 146.

“cooperate in completing eye examinations.”

4. Line 263, the reference to Supplemental Figure 2 first needs to occur in the results section. The discussion should spell out the implications of this finding that average AL/CRC changes with age. Presumably your threshold in this study is not going to be applicable to other age groups.

Thank you for your comment. We have revised as follows:

Page 8 Lines 197-198.

“Supplemental Figure 2 shows that AL/CRC ratio correlates with age, and the older the age, the greater the value of AL/CRC ratio.”

Page 12 Lines 270-271.

“This finding indicates that, in addition to refraction, age also affects the value of AL/CRC ratio.”

5. Supplemental Figure 2, thank you for adding this in and it does show that age appears to affect the average AL/CRC. My comment related to plotting the means data shown in Supplemental Table 3 which I think would more clearly show the age vs AL/CR relationship (given the larger age range), but I am satisfied with the authors current Supplemental Figure 2 if that is their preference.

Thank you for your comment. We think Supplemental Figure 2 shows a better visualization of the relationship between age and average AL/CRC.

VERSION 3 – REVIEW

REVIEWER	Lingham, Gareth Technological University Dublin
REVIEW RETURNED	13-Nov-2023
GENERAL COMMENTS	I thank the authors for addressing my comments. I ask that the authors rewrite the phrase: "Error lack of hyperopia reserve assessment" (line 229) to something like: "Using the optimal cut-off value of AL/CRC ratio, 22.22% (56/252) of participants with a lack of hyperopia reserve were not correctly classified." I have no other comments.

VERSION 3 – AUTHOR RESPONSE

Reviewer #5:

I thank the authors for their time in responding to my comments. However, I still have a few comments or suggestions for improvement remaining.

1. I ask that the authors rewrite the phrase: "Error lack of hyperopia reserve assessment" (line 229) to something like: "Using the optimal cut-off value of AL/CRC ratio, 22.22% (56/252) of participants with a lack of hyperopia reserve were not correctly classified."

Thank you for your comment. We have revised as follows:

Pages 10 Lines 225-226.

“Using the optimal cut-off value of AL/CRC ratio, 22.22% (56/252) of participants with a lack of hyperopia reserve were not correctly classified (Supplemental Table 3).”